# Graphene Synthesis: Method, Exfoliation Mechanism and Large-Scale Production

**Naixu Liu** [1,2] , **Qingguo Tang** [1,2,*], **Bin Huang** [1,2] **and Yaping Wang** [1,2,*]

[1] Key Laboratory of Special Functional Materials for Ecological Environment and Information, Hebei University of Technology, Ministry of Education, Tianjin 300130, China; liunaixu1996@gmail.com (N.L.); hb19931021@gmail.com (B.H.)

[2] Institute of Power Source and Ecomaterials Science, Hebei University of Technology, Tianjin 300130, China

\* Correspondence: tangqingguo@hebut.edu.cn (Q.T.); wangyaping@hebut.edu.cn (Y.W.)

**Abstract:** Graphene is a unique attractive material owing to its characteristic structure and excellent properties. To improve the preparation efficiency of graphene, reduce defects and costs, and meet the growing market demand, it is crucial to explore the improved and innovative production methods and process for graphene. This review summarizes recent advanced graphene synthesis methods including "bottom-up" and "top-down" processes, and their influence on the structure, cost, and preparation efficiency of graphene, as well as its peeling mechanism. The viability and practicality of preparing graphene using polymers peeling flake graphite or graphite filling polymer was discussed. Based on the comparative study, it is potential to mass produce graphene with large size and high quality using the viscoelasticity of polymers and their affinity to the graphite surface.

**Keywords:** graphene; preparation method; graphite; peeling mechanism; polymer material

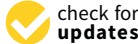



## 1. Introduction

Graphene is the general designation of single-layer, double-layer, and few-layer (3–10 layers) planar sheets comprising $sp^2$-bonded carbon atoms packed in a hexagonal honeycomb structure [1]. Ever since its discovery in 2004, graphene has been a focus in the research fields of materials, energy, and information owing to its excellent properties, such as high specific surface area (2630 $m^2$/g), high transmission rate (97.7%), high thermal conductivity (5000 W/mK), high Young modulus values (1.0 TPa), and high strength value (130 GPa). The electron mobility of graphene is $2 \times 10^5$ $cm^2$/vs, and it exhibits a significant room temperature quantum Hall effect [1–3]. Superconductivity occurs when two single-layer graphites are twisted to form double-layer graphene at 1.1 degrees [4]. Graphene can be combined with metals and oxides, compounds, or organic polymers to form a unique composite material [5–10], which has excellent application prospects in numerous fields such as supercapacitors [11], lithium-ion batteries [12], proton membrane fuel cells [13], and other energy conversion and storage materials, optoelectronic components [14], sensors [6], catalyst carriers [15], environmental functional materials [8], medical, and biological materials [16]. However, differences in the graphene preparation process not only affect the preparation cost and production efficiency of graphene, but also directly influence its number of layers, defect types, and surface properties, hence becoming an important factor restricting the large-scale application of graphene [1–4].

In this review, we present an updated overview focusing on the preparation methods and peeling mechanism of graphene, by highlighting the mechanism of exfoliation methods and the influence of different processes on the size and number of layers produced. In addition, the viability and practicality of preparing graphene using polymers peeling flake graphite was discussed.

## 2. Preparation Method of Graphene

Graphene preparation methods can be categorized into "bottom-up" construction method processes and "top-down" large-grain graphite crystal exfoliation methods [17].

The construction methods are based on the vaporization, pyrolysis, and reaction of low-boiling organics under high-energy effects such as the laser [18], microwave [19], plasma [20], pulsed laser deposition (PLD) [21–23], used to deposit or precipitate carbon atoms on the surface of the substrate. After the induced nucleation, growth, crystallization, or rearrangement, large-size single-layer and few-layer graphenes are finally obtained. According to the preparation processes, construction methods comprise chemical vapor deposition (CVD) [14,24], organic synthesis method [13,25], epitaxial growth method [26], in situ self-generating template method [15], and the high temperature and high-pressure growth method. CVD is an effective method for the large-area and high-quality preparation of single-layer graphene films [18,27]. For example, Wang Shuai's research group [24] achieved the controllable growth of centimeter-level single-crystal graphene by adjusting the ratio and concentration of methane, hydrogen, and oxygen, and found that [28] the oxidant is an effective regulator of graphene nucleation density and growth rate. By accurately controlling the $O_2$ concentration, the maximum growth rate of centimeter-level single-crystal graphene can reach 190 μm/min, which is about 475 times faster than the conventional 0.4 μm/s growth rate. CVD graphene is the mother element of graphene transistors, sensors, and transparent conductive films in the laboratory [14]. Figure 1 illustrates results of the as-grown graphene from different CVD processes.

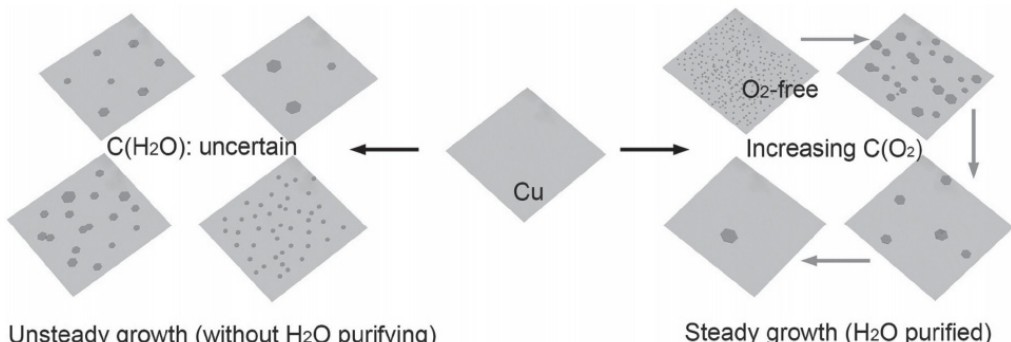

**Figure 1.** Illustration of grown graphene from different CVD processes. C ($H_2O$) and C ($O_2$) represent the concentration of $H_2O$ vapor impurities and introduced $O_2$, respectively [28]. Reprinted with permission from ref. [28]. Copyright 2016 WILEY-VCH Verlag GmbH & Co. KGaA.

CVD requires post growth processing to remove the catalyst (e.g., copper foil). A common transfer method is to use a polymer carrier, such as polymethylmethacrylate (PMMA). However, this technique is time-consuming, and requires specific equipment to achieve a high-quality transfer. Furthermore, the large-area graphene transferred via PMMA has impurities, defects, and, in some parts, noticeable cracks. Another approach to transfer graphene is by utilizing the thermal release tape (TRT). However, this process is also prone to leaving residue and is less effective for large transfers due to the inconsistent adhesion of the tape. Eric Auchter's research group [29] presented a novel method of transferring large-area graphene sheets onto a variety of substrates using Formvar (polyvinyl formal). The nature of the formvar dissolution allows graphene to be transferred onto virtually any substrate unaffected by chloroform, and requires only a one-minute immersion in chloroform to remove the sacrificial polymer. In the transfer area up to 3 cm$^2$, no folding or cracking on numerous graphene transfers via formvar was observed. By controlling the ratios of formvar, glycerol, and chloroform, Auchter et al. [30] synthesized tunable sub-micron-thick, porous membranes with 20–65% porosity; these formvar-based membranes had an elastic modulus of 7.8 GPa, a surface free energy of 50 mN·m$^{-1}$, and an average thickness of 125 nm. To reduce defects and impurities introduced during the transfer of graphene to the insulating substrate, Brzhezinskaya et al. [31] directly CVD synthesized

a few-layer graphene film on the insulating piezoelectric substrate $La_3Ga_{5.5}Ta_{0.5}O_{14}$ crystal. The step height between the substrate and graphene surfaces is approximately 1 nm, indicating that there are two or three graphene layers.

The mechanical, thermal, and electronic properties of polycrystalline graphene are much lower than those of single-crystal graphene [32], CVD can be used to effectively synthesize single-crystal, large-area, and monolayer graphene. Xu et al. [33] designed a temperature-gradient-driven annealing method to transform a polycrystalline Cu foil into a single-crystal Cu(111) foil. With this continuous annealing method, $5 \times 50$ cm$^2$ single-crystal Cu(111) substrates were obtained, and single-crystal monolayer graphene was grown on them. By restricting the initial growth temperature to between 1000 K and 1030 K, Wang et al. [34] produced large-area, fold-free, single-crystal monolayer graphene films on the $4 \times 7$ cm$^2$ single-crystal Cu–Ni(111) substrates using ethylene as the carbon precursor. However, the low yield, high preparation cost, high-energy consumption, and expensive equipment limits their application for mass production.

The exfoliation method, using flake graphite or pyrolytic graphite with high crystallinity as raw material, separates the stacked sheets through external forces such as impact, [35], shear [36], friction [37], airflow expansion [38], blasting [39], chemical intercalation [40], redox [41], and electrode reactions [42], which significantly weaken and destroy the Van der Waals forces between graphite sheets to form single-layer, double-layer, or few-layer graphene. According to the "power source" in the peeling process, it is categorized into electrochemical exfoliation [8,12,13,25–28], oxide-assisted liquid phase exfoliation [6,9,43], mechanical exfoliation [27,44–47], and three other categories.

Electrochemical liquid phase peeling is an effective method for the rapid preparation of few-layer graphene, carried out using a high-purity graphite rod as the cathode. During the electrochemical process, the cathode will gradually exfoliate and deposit on the anode. The ions in the electrolyte effectively prevent the exfoliated graphene sheets from re-aggregating. By combining this process with ultrasonic dispersion, few-layer graphene suspension with good dispersibility can be obtained [8,27,48]. However, the obtained graphene is mainly few-layer graphene or graphite nano-sheets, and the low peeling efficiency leads to the low quality of graphene.

The thermal reduction method is an effective approach to produce large amounts of graphene, which first oxidizes graphite using strong oxidants such as acid potassium permanganate and perchloric acid to obtain graphite oxide with carboxyl and hydroxyl groups on the edges, and epoxy and carbonyl groups between the layers. The distance between crystal planes is expanded to approximately 0.8 nm, and the Van der Waals force between graphite sheets is sharply reduced. After ultrasonic dispersion, stirring, grinding, and other mechanical forces, the single- or few-layer graphene oxide sheet is obtained. Then, reductants such as sodium borohydride, hydrazine hydrate, ascorbic acid, and dopamine are added to obtain graphene [49,50]. The reduction process includes microwave, electrochemistry, and the spontaneous reduction of the GO aerogel triggered by laser light [51]. This method can produce single-layer, double-layer, or few-layer graphene oxide and reduced graphene at low cost, which can be used as a functional component to meet the needs of materials, environmental, and chemical engineering. In practice, the high-energy consumption and complex process of the thermal reduction method limits its large-scale application. Meanwhile, the high consumption of oxidants, acids, and reductants, associated with the emission of toxic pollutants such as $NO_2$, $N_2O_4$, $HCl$, and $H_2SO_4$, will impede the green development of the graphene preparation industry [52]. Moreover, the damage to the edge of the graphene sheet and part of the lattice structure caused during the oxidation process is difficult to completely repair through the reduction process, and affects the electrical conductivity of the product to a certain extent.

To reduce the oxidation of graphite flakes and skeleton structure during the thermal reduction method, intercalation-aided exfoliation [8,12,46,53–55] can be used to promote the exfoliation of graphite. For example, Badri et al. [46] dispersed graphite powder in 0.5 mg/mL alkaline lignin aqueous solution to form a 4 mg/mL graphite suspension. After

heating intercalation and ultrasonic dispersion for 5 h, the graphite particles were removed by centrifugation to obtain a concentration of $0.72 \pm 0.05$ mg/mL low-defect and few-layer graphene suspension. Liu et al. [53] used a peroxyacetic acid and sulfuric acid mixture solution system to intercalate and exfoliate the graphite. The exfoliation mechanism and morphology of obtained graphene are shown in Figure 2.

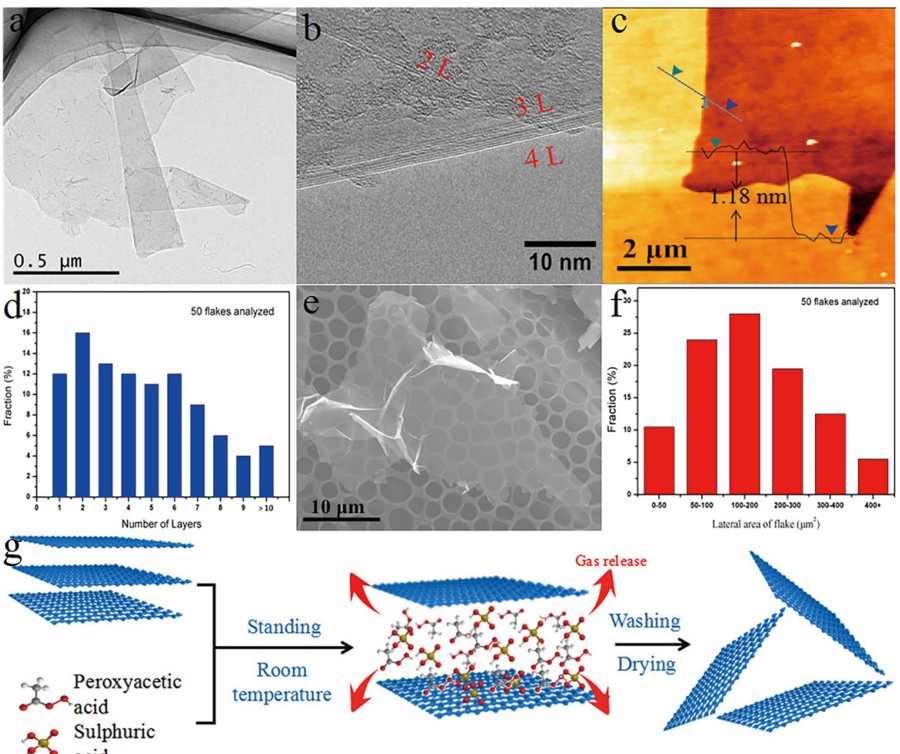

**Figure 2.** Characterization and intercalation mechanism of intercalated and stripped graphene. (**a**) Transmission electron microscopy (TEM); (**b**) High Resolution Transmission Electron Microscopes (HR-TEM); (**c**) Atomic force microscopy(AFM); (**d**) Thickness statistics of graphene with different layer numbers; (**e**) Scanning electron microscopy (SEM) image of graphene dispersed on TEM microgrid; (**f**) Lateral area distribution of graphene calculated from 50 flakes; (**g**) Schematic diagram of graphite intercalation and exfoliation [53]. Reprinted with permission from ref. [53]. Copyright 2018 Elsevier B.V.

Figure 2 shows the exfoliation product after peeling at room temperature for four hours. The yield of graphene is close to 100%, the average number of layers is less than five, and the maximum area can reach 420 $\mu m^2$. The intercalation agent method exhibits a good exfoliation effect as well. For example, Lin et al. [43] ground graphite flakes with ammonium bicarbonate ($NH_4HCO_3$), followed by heating at 60 °C under vacuum to intercalate $NH_4HCO_3$ between the graphite layers, followed by heating in a microwave to quickly decompose $NH_4HCO_3$ into $CO_2$, $NH_3$, and $H_2O$ vapor. The strong pressure generated instantly broke out to exfoliate the graphite sheet into few layers of graphene.

To increase the exfoliation speed of graphite and reduce graphene agglomeration, graphite can be dispersed into *N*-methylpyrrolidone (NMP), dimethyl sulfoxide (DMSO), *N*,*N*-dimethylformamide (DMF), and other different types of ionic liquid, followed by using a mechanical force to form few-layer graphene suspension, which is referred to as ionic liquid exfoliation [43,44]. Yousef et al. [43] prepared low-temperature expandable graphite (LT-EG)/DMF suspensions by a mechanical mixer unit, and used the second unit "multi-roller wet milling (MR-WM)" to synthesize as-prepared graphene from LT-EG/DMF under constant exfoliation for 120 min at speeds of 300 to 600 rpm. However, the high cost of ionic liquids, long reaction process, low yield, low preparation efficiency, and environmental

pollution emission severely restrict the development of ionic liquid exfoliation. To reduce costs and improve preparation efficiency, Gonzalez-Dominguez et al. [56] used a ball mill to mechanochemically intercalate organic molecules (melamine) into graphite, and washed suspended graphene to eliminate most of the melamine. The graphene products were three to four layers thick and ~500 nm in diameter on average, and most of the melamine could be removed by washing with water at 70 °C. Yoshihiko Arao et al. [57] used the high-speed laminar flow generated by a pressure homogenizer to effectively exfoliate large quantities of high-quality graphene; a production rate of 3.6 g/h of graphene in aqueous solution was achieved. Thus, effectively overcoming the Van der Waals forces between graphite layers is key to the efficient preparation of graphene.

Mechanical exfoliation is a traditional and simple method that has been applied for decades. However, the physical and chemical properties of graphene, such as the purity, size, number of layers, and performance are difficult to control, leading to the limited promotion value. For instance, the low efficiency of adhesive peeling results in low yield [1,17], which only meets the needs of laboratory research. In recent years, supercritical fluid (SCF) exfoliation [58], wet-jet milling exfoliation [59], and gas-driven exfoliation [60] have been developed, which are more efficient. Wang et al. [58] produced graphene at high-efficiency by supercritical $CO_2$ exfoliation with rapid expansion. Zhang et al. [60] used a high sheer rate of up to $3.3 \times 10^7$ s$^{-1}$ generated by driving high-speed gas at working pressures as low as 0.5 MPa to exfoliate graphite; the microscopic morphology and size changes of the stripped graphene are shown in Figure 3.

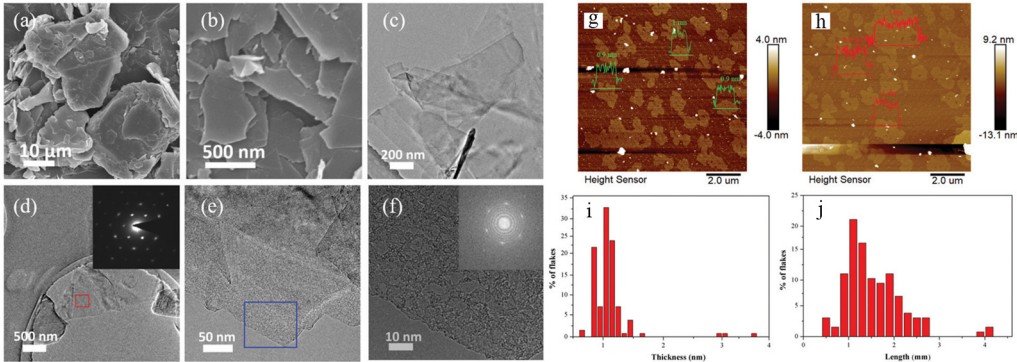

**Figure 3.** Microscopic morphology photos of gas-driven exfoliated graphene. SEM images of (**a**) graphite powder, (**b**) gas-driven exfoliated graphene. (**c**) Wide-field TEM image of gas-driven exfoliated graphene. (**d**,**e**) Representative monolayer flakes; inset in image (**d**) shows the selected area electron diffraction (SAED) pattern of the red-marked box. (**f**) HR-TEM image of blue-marked box in image (**e**); inset shows corresponding fast Fourier transform (FFT) pattern. (**g**,**h**) AFM images of gas-driven exfoliated graphene. (**i**) Thickness distribution of graphene. (**j**) Length distribution of graphene [60]. Reprinted with permission from ref. [60]. Copyright 2019 Royal Society of Chemistry.

Figure 3 shows that 62% of the obtained graphene sheets are single-layer, while 97% of flakes are ≤2 layers, with no lattice defect and high quality. However, the largest challenges of gas-driven exfoliation are that the number of graphene layers is difficult to control, the particle size is uneven, and the issue of "agglomeration" in grinding remains unsolved. From the perspective of the development of graphene preparation technology, novel processing methods continue to emerge, and numerous problems remain to be solved in large-scale preparation and utilization.

## 3. Exfoliation Mechanism of Graphene

Flake graphite crystal layers are connected by delocalized π bonds and Van der Waals forces, with binding forces of approximately 16.7 kJ/mol. Carbon–carbon atoms in graphite flakes are combined with sp$^2$ hybrid orbitals to form covalent bonds with a bond energy of up to 345 kJ/mol, more than 20 times greater than the bonding force

between layers [61]. The low interlayer bonding force is key to achieving the sliding, wrinkling, twisting, and peeling of graphite layers, and also provides the possibility for mechanical peeling[1] or chemical intercalation [46,53–55]. Zhang et al. [62] used a wet ball milling approach to synthesize SiC-graphene core-shell nanoparticles in situ from graphite and SiC nanoparticles. Graphite flakes were gradually exfoliated into fresh graphene nanosheets (GNSs) without significant defects, which is attributed to mechanical shearing and moderate impaction forces between graphite flakes, milling balls, and SiC nanoparticles during wet milling. It was estimated that >50% of the produced GNSs are wrapped around the SiC nanoparticles, and these GNSs are generally ≤6 layers. Based on the London–Van der Waals model, the required energy to exfoliate graphite with a thickness of 20 μm into single- and 100-layer graphite are $1.69 \times 10^{-11}$ J and $2.25 \times 10^{-11}$ J, respectively. Meanwhile, the moving $ZrO_2$ grinding ball with a diameter of 5.0 mm can transfer up to $2.77 \times 10^{-7}$ J of kinetic energy to the graphite, which is sufficient to exfoliate graphite to form single-layer graphene. Figure 4 shows two mechanisms for the formation of SiC-GNSs core-shell nanoparticles during wet ball milling.

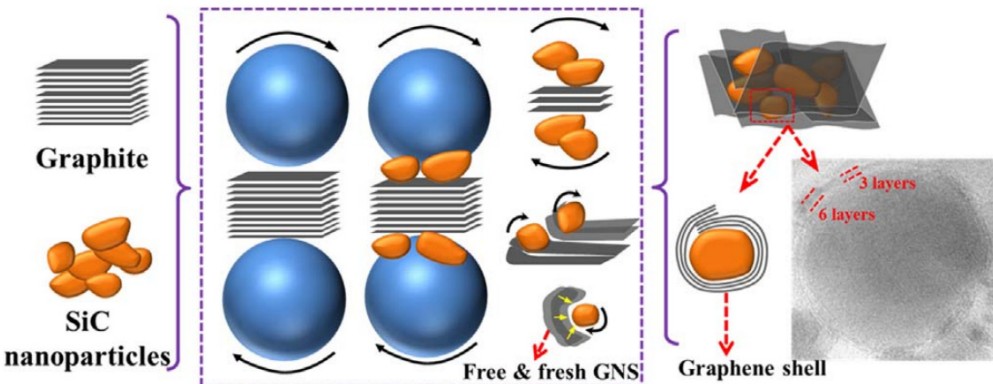

**Figure 4.** Schematic drawing showing exfoliation of graphite/GNSs and encapsulation of SiC nanoparticles with GNSs [62]. Reprinted with permission from ref. [62]. Copyright 2018 Elsevier Ltd. and Techna Group S.r.l.

In exfoliation theory and simulation research, Christian et al. [63] studied the theoretical model of the contact angle, surface energy, and graphite flake exfoliation by conducting a theoretical analysis of the crystal structure and thermodynamic definition of surface tension. Hamilton et al. [64] dispersed graphite powder in high-boiling o-dichlorobenzene (ODCB), and obtained graphene by ultrasonic peeling. The π–π interaction between ODCB and graphite flakes made the surface tension ($36.6 \times 10^{-3}$ J·m$^{-2}$) similar to the tension required for graphite peeling (($4–50) \times 10^{-3}$ J·m$^{-2}$), which explains the essence of ultrasonic-assisted peeling to form a few layers of graphene.

Yoon et al. [55] investigated the mechanism of intercalation-aided exfoliation of graphite using Van der Waals force-corrected density functional theory (DFT) calculations. Relative calculation results show that the required exfoliation energy of graphite varies significantly with the intercalation reagents, which is not only determined by molecular size of intercalation reagents, but also to the interaction force and bonding form between the intercalation agent and the graphite layer. The intercalation of negative or positive ions, such as $Li^+$, $K^+$, $F^-$, $Cl^-$, and $Br^-$ can increase the bonding force between the intercalating agent and the graphene layer, resulting in a 1.5–5 times higher exfoliation energy. This theory is in contrast to the general belief that intercalation agents can enlarge the interplanar spacing of graphite crystals and reduce the Van der Waals forces, thereby weakening the bonding force between graphene layers, which is important to the selection of intercalation agents and the improvement of exfoliation efficiency. Pan et al. [65] experimentally observed bending induced failure (local buckling) of multilayered graphene sheets (MLGSs) through an inverse blister test. Theoretical modeling and molecular dynamics simulations were conducted to investigate the bending behavior of MLGSs without

any edge constraint. They found that the exfoliation modes include interlayer shearing, rippling, and kink buckling/delamination. Exfoliation modes depend on the length and thickness of the graphite flake. By analytically deducing the bending stiffness of MLGSs before failure, the bending stiffness decreases dramatically below a critical length (~9.7 nm). The above research provides a theoretical foundation for the mechanical method of peeling off graphite to prepare graphene.

## 4. Large-Scale Exfoliation of Graphite Using Polymer Materials

Owing to the unique viscoelasticity [66] and high strength [67] of polymer materials such as rubber and resin, and the affinity [68] and wettability [69] between the carbon chain and carbon materials, graphite can be exfoliated into graphene using polymer materials. The high yield, rapid exfoliation rate, and low cost of this method provide it with commercial prospects. For example, Li et al. [7] dispersed graphite flakes into epoxy resin glue, and pressed them by three-rollers. The results reveal that graphite flakes were exfoliated in situ to an average aspect ratio of 300–1000, with a thickness of 5–17 nm few-layer graphene. Alessandro Aliprandi et al. [10] coated Persian beeswax diluted with glycerin on the rollers of a three-roller, and used the viscosity of beeswax and the tearing force of mechanical rotation to peel off graphite flakes. The obtained product is few-layer graphene mainly with 4–5 layers and a thickness of ~2.6 nm.

In our study, graphene was successfully prepared by blending natural flake graphite with neoprene and natural rubber [70], followed by the compression, friction, and shearing of the roller machine. The detailed process is described as follows. First, rubber was mixed with pre-treated graphite in a two-roll mill at 60 °C for 90 min, a composite rubber block containing 10% graphene is finally formed. Second, a 1-g graphene composite rubber block was placed in the reaction kettle, and 50–100 mL chloroform or benzene solvent was added. After the hydrothermal process at 160 °C for 8–12 h by shaking the reaction kettle every 1 h, the graphene rubber suspension was obtained. Third, the graphene rubber suspension was centrifuged for 10 min at 2000 rpm to remove incompletely dissolved colloidal particles and incompletely peeled graphite particles in the suspension. The obtained a rubber suspension containing a certain concentration of graphene was then precipitated by a centrifuge at 10,000 rpm for 15–20 min. After washing with absolute ethanol, centrifuging, and freeze vacuum drying, the fluffy graphene powder was finally obtained.

Successful exfoliation of graphite was confirmed by microscopy and X-ray diffraction analyses, and the results are shown in Figure 5. Figure 5a shows that the obtained graphene presents a uniform flake-like structure with a size of approximately 2 μm. The edges are mostly piled together in a curled or folded form; the TEM image (Figure 5b) shows that graphenes are superimposed with a jagged structure on the edges. The atomic force microscopy (AFM) image reveals graphene sheets with numerous notches at the periphery. The average thickness of graphene is approximately 2.1 nm, indicating that there are three graphene layers.

The X-ray diffraction (XRD) analysis of Figure 5d shows that the diffraction peak intensity of natural flake graphite crystals at 2θ of 26.7° is 34,072.8, while the diffraction peak intensity of the graphene stacked flakes formed after exfoliation is only 786.3. The peak half-width is significantly larger, indicating that most graphite flakes have been dissociated, and graphene was formed [56]. Even if a few graphite flakes have not been completely peeled off, it does not have significant impact on the performance and application of graphene. Therefore, the rubber mixing process system can effectively exfoliate the flake graphite crystals into single-layer or few-layer graphene. Meanwhile, the great reproducibility and high yield of our study award it with commercial prospects.

Compared with various preparation methods of graphene, the utilization of polymer materials as the peeling agent is simple, scalable, and low-cost. Hence, we believe that it provides a novel top-down approach for the preparation of high-quality graphene on a large scale.

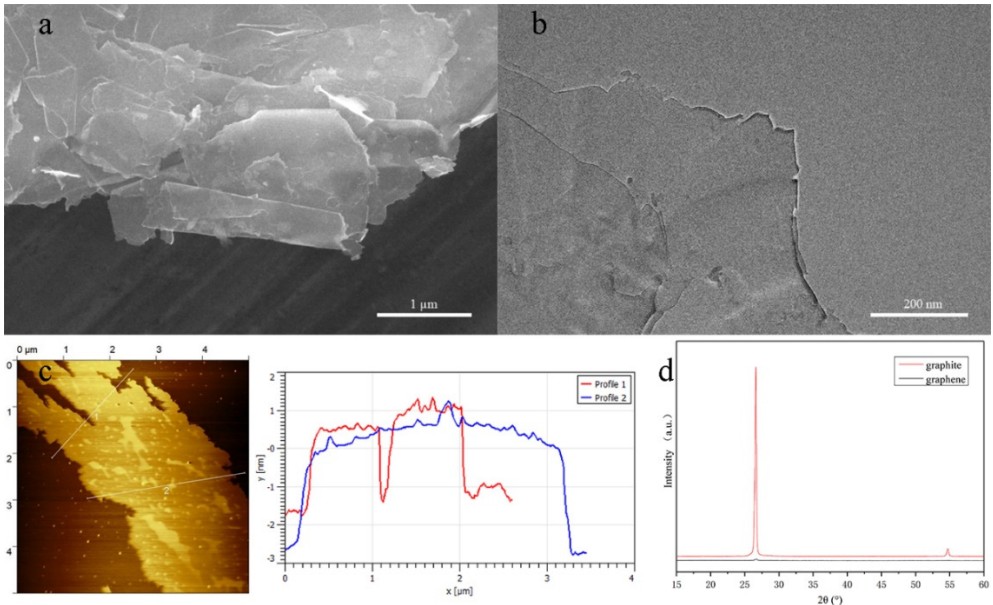

**Figure 5.** Microscopic morphology photos of rubber exfoliated graphene. (**a**) SEM; (**b**) TEM; (**c**) AFM image and graphene thickness change curve; (**d**) X-ray diffraction (XRD) analysis curve of graphite, and exfoliated graphene [70]. Reprinted with permission from ref. [70]. Copyright 2019 UNIV HEBEI TECHNOLOGY, 201910640615.3, China patent.

## 5. Summary and Outlook

Graphene preparation methods are constantly updated with the emergence of novel preparation processes and continuous in-depth study of the exfoliation mechanism. In this review, recent advances in the preparation methods are addressed. "Bottom-up" methods yield high-quality and large-size graphene. However, bottom-up approaches generally suffer from high production cost, complex processes, low efficiency, and limited production. The top-down exfoliation method requires simple equipment, but the content, size, number of layers, and defect types of prepared graphene are difficult to control. Preparing graphene using polymer materials, such as rubber and resin, as binders has advantages including simple processing, continuous and controllable scale, high production efficiency, less product defects, and wide application. Therefore, we believe that the polymer stripping graphene method is an important direction for the preparation of graphene and its composites.

**Author Contributions:** Writing—original draft preparation, N.L.; investigation, B.H.; writing—review and editing, Q.T.; supervision, Y.W. All authors have read and agreed to the published version of the manuscript.

**Funding:** This research was funded by the National Key Research and Development Program of China, grant number 2019YFC1904601.

**Data Availability Statement:** The data that support the findings of this study are available from the corresponding author upon reasonable request.

**Conflicts of Interest:** The authors have no conflict to disclose.

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
