# Peer review of "Graphene Synthesis: Method, Exfoliation Mechanism and Large-Scale Production"

_crystals, doi:10.3390/cryst12010025_

Round 1

Reviewer 1 Report

The manuscript entitled "Graphene Synthesis: Method, Exfoliation Mechanism and Large-Scale Production" deals with the graphene synthesis methods, especially with the top-down exfoliation methods by emphasing a novel method using a polymer material as binders. The idea of comparing this novel method with the others is relevant and this synthesis technique may be an alternative to the other top-down method.  Therefore,  I can only recommend the manuscript for publication after a minor revision is done addressing the points below:

Page 1: Line 12-13: "This review summarizes recent advanced graphene synthesis methods including "bottom-up" and "top-down" processes, and their influence on the number of layers, particle size, surface defects or structure, cost,  and preparation efficiency of graphene, as well as its peeling mechanism."

This sentence is a bit confusing because in the text I did not see any discussion about the number of layers, particle size, surface defects of other graphene growth methods except the exfoliation methods. 
This sentence need to be rewrite to match with the body of the review.

Page 2: Line 49-50:

Other method like PLD pulsed laser deposition can also be used for graphene synthesis. Can you please add this method in the list you quote in line 49-50.
You can consider for example the following papers:
1- Review of graphene growth from a solid carbon source by pulsed laser deposition (PLD)
2-Graphene synthesis on SiO2 using pulsed laser deposition with bilayer predominance
3-Boron doped graphene synthesis using pulsed laser deposition and its electrochemical characterization

Page 8: Line 304-305: "However, bottom-up approaches generally suffer from high production cost, complex processes, low efficiency, and limited production."

The authors did not develop this statement in the body of the review.
I recommand more comments on this statement  in the body of the review by just giving an example with some references.

Author Response

Response to Reviewer 1 Comments

Point 1:

Page 1: Line 12-13: "This review summarizes recent advanced graphene synthesis methods including "bottom-up" and "top-down" processes, and their influence on the number of layers, particle size, surface defects or structure, cost, and preparation efficiency of graphene, as well as its peeling mechanism."

This sentence is a bit confusing because in the text I did not see any discussion about the number of layers, particle size, surface defects of other graphene growth methods except the exfoliation methods. 
This sentence need to be rewrite to match with the body of the review.

Response 1:

Thanks for your comments. This sentence has been rewritten as" This review summarizes recent advanced graphene synthesis methods including "bottom-up" and "top-down" processes, and their influence on the structure, cost, and preparation efficiency of graphene, as well as its peeling mechanism." on Page 1: Line 12-13. After deleting the part you suggest, I think it may match the body of the comment.

Point 2:

Page 2: Line 49-50:

Other method like PLD pulsed laser deposition can also be used for graphene synthesis. Can you please add this method in the list you quote in line 49-50.
You can consider for example the following papers:
1- Review of graphene growth from a solid carbon source by pulsed laser deposition (PLD)
2-Graphene synthesis on SiO2 using pulsed laser deposition with bilayer predominance
3-Boron doped graphene synthesis using pulsed laser deposition and its electrochemical characterization

Response 2:

Thanks for your comments. The PLD pulsed laser deposition method has been added on Page 2: Line 49-50, and three references you suggested have been cited.

Point 3:

Page 8: Line 304-305: "However, bottom-up approaches generally suffer from high production cost, complex processes, low efficiency, and limited production."

The authors did not develop this statement in the body of the review.
I recommand more comments on this statement in the body of the review by just giving an example with some references.

Response 3:

Thanks for your comments. I have added comments on this statement on Page 3: Line 91-92, and cited three references to illustrate.

Reviewer 2 Report

Reviewer report on manuscript Crystals-1515548

Liua et al. “Graphene Synthesis: Method, Exfoliation Mechanism and Large-Scale Production”

This review summarizes recent advanced graphene synthesis methods including "bottom-up" and "top-down" processes, and their influence on the number of layers, particle size, surface defects or structure, cost, and preparation efficiency of graphene, as well as its peeling mechanism. The viability and practicality of preparing graphene using polymers peeling flake graphite or graphite filling polymer was discussed.

The manuscript can be accepted after minor revision.  Authors should make the following corrections:

  1. In the section 4 “Large-scale exfoliation of graphite using polymer materials” (page 8), Line 279: “spectroscopic” should be replaced by “X-ray diffraction”.
  2. In the section 2 “Preparation method of graphene” (pages 1-5): the information about the method of direct growth of graphene film on piezoelectric La3Ga5Ta0.5O14 crystal should be added with reference to work “Physica Status Solidi (RRL), 2016, 10(8), 639”.

Author Response

Point 1:

  1. In the section 4 “Large-scale exfoliation of graphite using polymer materials” (page 8), Line 279: “spectroscopic” should be replaced by “X-ray diffraction”.

Response 1:

Thanks for your comments. The “spectroscopic” has been replaced by “X-ray diffraction” in the section 4 “Large-scale exfoliation of graphite using polymer materials” (page 8), Line 295.

Point 2:

  1. In the section 2 “Preparation method of graphene” (pages 1-5): the information about the method of direct growth of graphene film on piezoelectric La3Ga5Ta0.5O14crystal should be added with reference to work “Physica Status Solidi (RRL), 2016, 10(8), 639”.

Response 2:

Thanks for your comments. I have added the information about the method of “direct growth of graphene film on piezoelectric La3Ga5Ta0.5O14 crystal” on page 2, Line 86, and the reference you suggested has been cited.